# Development and Initial Validation of the Oral Health Activities Questionnaire

**DOI:** 10.3390/ijerph19095556

**Published:** 2022-05-03

**Authors:** Diana Aranza, Alessandro Nota, Tea Galić, Slavica Kozina, Simona Tecco, Tina Poklepović Peričić, Boris Milavić

**Affiliations:** 1Department of Health Studies, University of Split, 21000 Split, Croatia; 2Dental School, IRCCS San Raffaele Hospital, Vita-Salute San Raffaele University, 20132 Milan, Italy; nota.alessandro@hsr.it (A.N.); tecco.simona@hsr.it (S.T.); 3Department of Prosthodontics, Study of Dental Medicine, School of Medicine, University of Split, 21000 Split, Croatia; tea.galic@mefst.hr (T.G.); tina.pokepovic@mefst.hr (T.P.P.); 4Department of Psychological Medicine, School of Medicine, University of Split, 21000 Split, Croatia; slavica.kozina@mefst.hr; 5Faculty of Kinesiology, University of Split, 21000 Split, Croatia; boris.milavic@kifst.hr

**Keywords:** oral hygiene, oral health status, orientation to DMD, regularity of tooth brushing, toothache, tooth fillings, university students

## Abstract

*Background:* The purpose of this study was to introduce a new Oral Health Activities Questionnaire (OHAQ, hereinafter) that examines different activities and behaviours related to the oral hygiene regimen of each analysed subject. *Methods:* A sample of 658 students was analysed to determine the OHAQ scale’s basic metric characteristics. To determine the construct validity of the OHAQ, descriptive statistics and correlation analysis, as well as differences testing, were applied to groups of subjects on the basis of *self-reported oral status* measures. *Results:* The dimensions of oral health activities were determined, and the scales for their measurement were constructed. Females and males differed in the OHAQ questionnaire measures. Significant but low intercorrelations were found among the measures. In the female and male subsample, four different *oral health* (OH, hereinafter) *types* of subjects were identified, exhibiting different characteristic behaviours regarding oral health. OHAQ scales showed good *discriminant validity*, revealing the differences related to specific *self-reported oral status* measures (e.g., *frequency of toothache* and *the number of filled teeth*). *Conclusions:* The OHAQ represents a satisfactory measurement instrument for determining the level of OH activities and for doing quick and reliable classifications of the participating subjects according to their OH activities and behaviours. The process of further validation and advancements of the OHAQ scales and measures should be continued through a clinical examination of subjects.

## 1. Introduction

Oral health is an important part of overall health [1]. Different factors, such as lifestyle, habits, diet, frequency of dental check-ups, and socioeconomic status, affect the oral health of an individual [2,3,4]. According to the First International Conference on Health Promotion held by the World Health Organisation (Ottawa, Canada, 1986), oral health promotion is a combination of health education, healthcare, and health policies that aim to advance the oral health of the general population. Caries and periodontal disease are highly behaviour-related and can be controlled by proper oral hygiene activities [5].

Appropriate oral hygiene implies the continuous implementation of two well-defined sets of behaviour: self-protection (oral hygiene, fluoride usage, and reduced intake of refined carbohydrates), and regular utilisation of dental services (oral health education, regular dental check-ups, and professional prophylaxis) [6].

The most effective and widespread method of oral hygiene is toothbrushing. Oral self-care, including toothbrushing and interdental cleaning, is important for preserving oral health and preventing oral diseases, because it disrupts and removes microbial plaque, thus preventing its accumulation on the teeth and the gingiva [7]. Inappropriate toothbrushing techniques may be ineffective in plaque removal and even cause hard tissue abrasions or gingival recessions [8]. Therefore, knowledge about oral hygiene, including the products, procedures, and behaviours, is an important factor in preventing oral diseases and achieving good oral health [8,9,10].

Different social factors, such as level of education, employment status, and work conditions, as well as other health-related habits, have also been shown to affect oral health. The improved oral health of regular dental patients seems to be more affected by professional care level than the patient’s knowledge about oral health [11]. Previous studies have shown that oral hygiene habits and attitudes are gender-related, with females having better dental health attitudes and behaviours [12].

Factors that influence the effectiveness and adequacy of the patients’ oral hygiene level include their knowledge, attitudes, and behaviour regarding oral disease prevention. Multiple studies compared oral health attitudes and behaviour in different countries among students, and most of them used the Hiroshima University Dental Behavioural Inventory (HU-DBI), developed by Kawamura [13,14,15,16,17,18]. This inventory consists of 20 dichotomous questions (agree/disagree) and aims to investigate the behaviour of patients during toothbrushing to predict their clinical outcomes. However, despite the widespread utilisation of this inventory, to date, no other tool has been developed to evaluate behaviours related to all the other oral health activities, such as dental flossing, interdental brushing, or the choice of toothpaste. Numerous studies used the HU-DBI questionnaire to investigate gender differences in oral health-related knowledge and behaviour among dental students worldwide [13,14,15,16,19,20]. Evidence on the oral health status in other population groups, such as children or older adults, is often based on national or regional epidemiological studies that employed the DMFT (decayed, missing, and filled teeth) index [21,22,23]. This study suggests the use of the Oral Health Activities Questionnaire (OHAQ) to broaden the available scientific findings and fill literature gaps by including students from various faculties.

Therefore, the purpose of this study was to develop and initially validate the OHAQ intended to identify oral hygiene-related activities and behaviours, in addition to the level of oral self-care within the population of university students, as well as to serve as a screening tool to detect individuals that might require immediate dental treatment.

## 2. Materials and Methods

### 2.1. Participants

A total of 658 students from the University of Split (Split, Croatia) were included in the study, 439 (66.7%) women and 219 (33.3%) men, whose mean age was 21.33 ± 2.61 years (age range 18 to 26 years). To calculate the minimum sample size, the Raosoft sample size calculator was used (Raosoft Sample Size Calculator). Therefore, with the estimated university student population size of 20,000, confidence level of 99%, margin of error of 5%, and response distribution of about 50%, the minimum effective sample size calculated for this investigation was 643. Representation of students from different faculties was as follows: Faculty of Economics (90 students, 13.7%), Faculty of Electrical Engineering, Mechanical Engineering and Naval Architecture (75 students, 11.4%), Faculty of Philosophy (78 students, 11.9%), Faculty of Kinesiology (53 students, 8.1%), School of Medicine (86 students, 13.1%), School of Dental Medicine (90 students, 13.7%), Faculty of Law (100 students, 15.2%), Faculty of Science (46 students, 7%), and the University Department of Health Studies (40 students, 6.1%). The students at the University of Split were invited to participate in this study at the beginning of the spring semester.

### 2.2. Development of the Oral Health Activities Questionnaire (OHAQ)

The OHAQ items were structured by first creating a wide pool of items related to oral hygiene and oral health practices. The items either corresponded with usual and expected behaviour or represented specific activities as the gold standard of oral hygiene. Initially, three dental medicine doctors (DMDs) with more than 15 years of clinical experience constructed the items of manifest behaviour and important oral hygiene activities in the form of statements or claims. The DMD expert group assessed the *content validity* of the OHAQ items to determine the unbiased relevance of the items to the overall oral health construct and the items’ ability to measure a specific oral hygiene activity using simple and easy-to-understand terminology. All items were given to a group of 25 nursing university students to rate each item according to two criteria: clarity and applicability. Those students were not included in the study sample. The items that were marked as unclear or not applicable were additionally revised by the team or excluded from further consideration. Each item was evaluated using a five-point Likert scale (1—*completely false*; 2—*mostly false*; 3—*partially true*; 4—*mostly true*; 5—*completely true*).

### 2.3. Self-Reported Oral Health Status Questionnaire

In addition to the OHAQ, all participants were asked to objectively evaluate their actual *oral health status* by providing assessments on a few additional questions. These additional questions were later used to evaluate the initial validity of the OHAQ, assuming that their actual oral health status, self-reported by the participants, was a consequence of their previous and presently applied oral health activities and behaviours. The questions used to determine the *self-reported oral health* status were divided into two groups. The first group consisted of questions in which respondents assessed the condition of their teeth, but also their experience in maintaining oral care: *filled teeth*, *tooth extraction*, *root canal treatment*, *malocclusion*, *prosthodontic treatment*, *orthodontic treatment*, and *dental crowns or veneers*, by providing just *“**yes*” or “*no*” answers. The respondents who answered “yes” were also asked to enter a numerical value (e.g., the number of *filled teeth* regardless of the reason for the specific filling) for those questions. In questions for which the respondents entered numerical value (questions about the number of *filled teeth*, the number of *extracted teeth*, the number of *root canal-treated teeth*), the entered number served as data in the database. For those who did not have such experiences, 0 (*zero value*) was entered. For questions to which the respondents could only give an affirmative or negative answer (*malocclusion*, *prosthetic treatment*, *orthodontic treatment*, and *dental crowns or veneers*), the values 0—*no experience* or 1—*affirmative*, *has experience* were entered in the database. The second group consisted of three questions concerning *toothache frequency*, the frequency of *use of analgesics*, and the frequency of *use of antibiotics* for dental reasons. Respondents rated them on a Likert scale with the answers offered: 1—*never*, 2—*very rare*, 3—*rare*, 4—*sometimes*, and 5—*often*. We assumed that a higher incidence in all these questions (e.g., higher number of *filled teeth* or more frequent *toothache*) could objectively represent poor and less desirable characteristics of individuals’ oral health status, as possible consequences of either previous or current lack or frequent improper application of adequate oral health activities.

### 2.4. Application of the OHAQ on a Sample of University Students

The study was conducted in full accordance with the World Medical Association Declaration of Helsinki and approved by the Ethics Committee of the University of Split, Department of Health Studies. The authors contacted the faculty of the University of Split to enlist potential participants. The questionnaire was applied in regular class groups, and students were asked by their faculty to remain in class at the end of a lecture to participate in this study voluntarily. The aim and purpose of the study were explained to the students, and they were given instructions on how to fill out the questionnaire. All of the students were asked to complete the OHAQ anonymously after having signed a separate consent form for participation in the study. The questionnaire was broader than the described variables in this study, and students filled it out in 15–20 min. As the students participated voluntarily, a very small number of the questionnaires (less than 1% of the overall sample) were excluded from the analyses at the end of the study, mainly because the participants did not answer or unclearly answered several items in the questionnaire.

### 2.5. Data Analysis

To determine the structure of oral health activities, *principal component analysis* was applied to the initial set of OHAQ items using *Varimax* orthogonal rotation and the *Kaiser–Guttmann* criterion. This approach helped with the extraction of significant components. Afterward, separate subscales were constructed for each of the yielded OHAQ dimensions. Proper procedures for determining their metric characteristics (*homogeneity*, *reliability*, and *sensitivity*) were applied. Their internal consistency was assessed by calculating Cronbach’s *alpha* coefficient. Several *sensitivity* indices were calculated: measures of dispersion (minimum and maximum values of the scale and median of results), the coefficient of the Kolmogorov–Smirnov *goodness-of-fit* test, and measures of distribution (*skewness* and *kurtosis*). Descriptive parameters were calculated for all OHAQ measures—scales on the total sample (subsamples). Continuous data were presented as the *means ± standard deviations* (*mean ± SD*) and median values. The data in the questions representing the oral status of the subjects were also analysed, whereas categorical variables were presented as observed frequencies and relative percentages. Correlation analysis was performed to determine the association between the OHAQ measures. Later, subjects from the sample were categorised into particular groups according to the number or the frequency values in some variables. Differences tests (Student’s *t*-test; Fisher’s *post hoc least significant difference* test, LSD; *one-way ANOVA*) were then applied to analyse the differences between the *OHAQ groups* and the *self-reported oral health status* groups of subjects. The *K-means clustering* method was used to determine the subjects’ *oral health types* measured by the OHAQ questionnaire, with the number of clusters determined a priori. In determining the cluster memberships, the option with the smallest possible differences between members within a cluster and with the largest possible differences between different clusters was used, regardless of the frequency of members in each cluster. Finally, a *chi-square test of association* was used to calculate the correlation of two category variables and *Cramer’s V coefficient* was included in the analysis as an *effect size* index of the *chi-square test*. In all conducted statistical analyses, the lowest significance criterion was set at *p* < 0.05. Statistical analyses were performed using the statistical software package Statistica 14 (StatSoft Inc., Tulsa, OK, USA).

## 3. Results

*Principal component analysis* yielded five dimensions of oral health activities explaining the 51.1% of the total variance (Table 1).

These five dimensions related to the oral health activities questionnaire were the following: the first dimension, named *basic oral hygiene activities* (BOHA, hereinafter), involved six items that described the basic knowledge, manner, time, and instruments used by participants, or basic manifest behaviours when practising oral hygiene; the second dimension, named *orientation to dental medicine doctor* (ODMD, hereinafter), involved four items that described the regularity of dental scaling and dental check-ups; the third dimension, named *regularity of tooth brushing* (ROTB, hereinafter), involved three items that described when and how often patients brushed their teeth; the fourth dimension, named *use of dental floss* (FLOSS, hereinafter), involved two items; the fifth dimension, named *additional and detail oral hygiene activities* (ADOH, hereinafter), involved four items that described the use of additional oral hygiene products. Each component explained from approximately 8.7% to 12.0% of the variance of the questionnaire, and the *basic oral hygiene activities* component explained the greatest portion of the total variance.

Table 2 reports the basic metric characteristics of each OHAQ subscale and the descriptive characteristics of each item.

As seen in Table 2, four of the five scales showed *satisfactory* metric characteristics for reliability and homogeneity: the *basic oral hygiene activities* scale, the *orientation to a doctor of dental medicine* scale, the *regularity of tooth brushing* scale, and the *use of dental floss* scale. These scales showed good *homogeneity* because all items of each scale were related to a single component, and their internal consistency coefficient (Cronbach’s α) varied from *conditionally satisfactory* (0.64) to *good* (0.84). In contrast, the *detailed oral health activities* scale showed a low and *unsatisfactory* level of internal consistency (Cronbach’s α: 0.42) and was excluded from further analyses.

The *sensitivity* of the scales, including the calculated sum of the OHAQ scales, was tested using the Kolmogorov–Smirnov test and coefficients of skewness and kurtosis to assess the normality of data distribution. Gender differences between the female and male subsample were calculated for each subscale. These data on *sensitivity* and gender differences are presented in Table 3.

The data confirmed the satisfactory *sensitivity* of the scales, as the values of skewness and kurtosis did not exceed the level of ±1.00, although some of the Kolmogorov–Smirnov test coefficients were significant. Therefore, the authors assumed that this distribution allowed for the application of parametric statistical procedures to the results of the scales, with the expectation of a high level of probability that there would be no significant violations of the basic assumptions of parametric statistics. The results revealed statistically significant gender differences, as females obtained higher and more desirable results than the males in all the scales. The differences were more evident in the measures concerning the *regularity of tooth brushing* and *basic oral hygiene activities*.

Statistically significant intercorrelations were found among the measures in all four scales and both gender subsamples (Table 4). Correlations between the OHAQ scales and the sum OHAQ measure were positive and high, ranging from 0.64 to 0.77.

Table 5 shows the *oral health types* (OH types, hereinafter) of respondents by their expression of OHAQ measures, calculated separately for both gender subsamples. Within the subsamples of female and male respondents, all identified *OH types* differed significantly in the degree to which they were assessed on the OHAQ scales.

Four types of OH were identified in the *female* subsample:The first type, which included 19.6% of female students, was called the *excellent OH type* because their scores on all OHAQ scales were *very high*;The second type, which included 27.3% of female students, was identified as the *good OH type* without flossing because their scores on the three OHAQ scales were *approximately high* and the flossing score was *extremely low*;The third type, which included 27.3% of female students, was called the *satisfactory OH type* because their scores on all four OHAQ scales were *approximately moderate*;The fourth type, which included 25.7% of female students, was called the *poor OH type* because their assessments of the three OHAQ scales were *low* and the assessment of flossing was *extremely low*.

Four types of OH were identified in the male subsample:The first type, which included only 9.1% of male students, was called the *excellent OH type* because their scores on all OHAQ scales were *very high*;The second type, which included 32.4% of male students, was called the *good OH type* without flossing because their ratings on the three OHAQ scales were *approximately satisfactory* and the flossing score was *extremely low*;The third type, which included 30.1% of male students, was called the *satisfactory OH type* because their assessments on all four OHAQ scales were *moderate*;The fourth type, which included 28.3% of male students, was referred to as the *poor OH type* because their scores on three OHAQ scales were *low* and their assessment of flossing was *extremely low.*

The differences between the OH types for female and male students are presented in Figure 1. Although the OH types of respondents in both gender subsamples had the same names and similar structures, it is noticeable that the OHAQ measures in the female subsample were somewhat more pronounced.

Descriptive statistics of *self-reported oral status* measures are shown in Table 6 in three separate sections.

The first part of the table refers to the total number of teeth extracted and treated and the number of teeth with fillings. Although male and female students had, on average, <1 extracted tooth, <3 filled teeth, and <2 treated teeth, the measures of dispersion of these variables were high. This indicates a low frequency of students with a high number of extracted teeth, treated teeth, and teeth with fillings. The second part deals with questions about the respondents’ assessment of the frequency of *toothache* and the frequency of using *analgesics* and *antibiotics* for toothache, and both female and male students gave a low self-assessment on these variables, mostly *very rarely* or even less frequently. The third part refers to four questions in which respondents answered whether they had experienced *malocclusion* (F = 27.8%; M = 26%), *orthodontic treatment* (F = 45.6%; M = 37.4%), *dental crowns or veneers* (F = 5.7%; M = 5.5%), or *prosthetic*
*treatment* (F = 2.3%; M = 2.7%).

One-way ANOVA (Table 7) was used to determine the differences between certain *OH types* of subjects in relation to the *self-reported oral status* variables. Variables *prosthetic*
*treatment* and *dental crowns or veneers* were not included in this analysis due to very low incidence in the subsamples.

For the female subsample, significant differences were found in two oral status variables: the *number of filled teeth* and *frequency of toothache* (*p* < 0.01). No significant differences were found in the male subsample, but the results of the analysis for three variables were very close to the criterion of significant difference. For two *self-reported* oral status variables (*malocclusion* and *orthodontic treatment*), additional nonparametric tests (*Kruskal–Wallis* test and *median* test) were performed to analyse the differences, and it was found that ANOVA and the nonparametric tests gave identical results.

Fisher’s *least significant difference* (LSD) *post hoc* test was applied for some self-reported oral status variables. It was found that, in the female subsample, in the variable *number of filled teeth*, the *excellent* OH type differed significantly from the *satisfactory* and *poor* OH types, with females with *excellent* OH having more teeth with fillings than the other two OH types. Moreover, the *good* OH type differed from the *poor* OH type, with females with *good* OH having more teeth with fillings than the *poor* OH type. In the female subsample, the *toothache* frequency variable distinguished the *poor* OH type from all other OH types, with females with *poor* OH having the most frequent *toothache*.

In the malesubsample, the variable *toothache* was found to distinguish the *excellent* OH type from all other OH types, with *excellent* OH males being the least likely to have *toothache*. Furthermore, in male subjects, the *use of*
*analgesics* variable distinguished the *satisfactory* OH type from all other types of OH, with men with *satisfactory* OH most frequently *using analgesics for*
*toothache*. In the *use of antibiotics* variable, the *satisfactory* OH type differed from the *good* OH type, with men with *satisfactory* OH using *antibiotics* more frequently than men with *good* OH. 

Since it was found that the different OH types of female and male respondents differed most in variables *toothache* and *the number of*
*filled teeth*, additional analysis was performed for these two variables. These two variables were set as *dependent variables*, and a *one-way ANOVA* analysis of the differences was performed between different *toothache* and *the number of filled teeth* groups of subjects. Because of the very low frequency of specific groups, some groups for comparison were formed from several different outcomes (e.g., *sometimes + often* group for *toothache*, and *six or more fills* group for *the number of*
*filled teeth*). The results of the analysis are presented in Table 8 for the female subsample and Table 9 for the male subsample.

Female respondents differed significantly in all OHAQ measures in terms of *toothache* frequency and in three of four OHAQ measures in terms of the number of *filled teeth* (except for the variable *regularity of tooth brushing*). It was observed that females who *never* experienced tooth pain practised such *basic activities of oral hygiene* more frequently and better than females who *sometimes/often* or *rarely* experienced tooth pain. Moreover, it was observed that females who *very rarely* experienced tooth pain practised such *basic activities of oral hygiene* more frequently and better than females who *rarely* experienced tooth pain. Regarding *orientation to a DMD*, females who experienced *toothache sometimes/often* differed from the others, as shown by lower values.

Furthermore, according to the *regularity of tooth brushing*, females who *never* experienced *toothache* differed from those who *sometimes/often* or *rarely* experienced tooth pain, as shown by their higher values. Moreover, it was observed that females who *very rarely* experienced *toothache* reported *regularity of tooth brushing* higher than females who *rarely* experienced tooth pain. According to the *use of dental floss*, females who *never* experienced *toothache* differed from the others, and they used dental floss *more often*. Females with *no* or *one filled tooth* differed from the others in the *lower* level of basic oral hygiene activities; females with *six or more filled teeth* differed from the others in the more frequent *use of dental floss* and a higher level of *orientation to a DMD*.

Regarding the frequency of *toothache* among the female respondents, 25.7% of them answered that they *never had toothache*, and just 8.7% of them *sometimes or often* had *toothache*. The frequency of the number of *filled teeth* among females shows that 38.7% had *no or only one* filling in their teeth, and that 11.9% had *six or more fillings.*

Male subjects differed significantly in the OHAQ measure of *toothache* frequency in relation to the *basic oral hygiene activities* (BOHA) variable, but did not differ in relation to the number of *filled teeth* (Table 9).

According to *basic oral hygiene activities*, males who *never* experienced *toothache* differed from the others in their higher level of basic oral hygiene activities. Furthermore, according to the *regularity of tooth brushing*, males who *never* experienced *toothache* differed from those who have *rarely* experienced *toothache*, as shown by their higher values. Additionally, males with *two* or *three filled teeth* differed from those with *no* or *one filled tooth* and those with *six or more filled teeth* by their *higher* level of *basic oral health activities*. The frequency of *toothache* in male subjects showed that 30.6% of subjects *never had toothache*, and 6.4% had it *sometimes or often*. Regarding the frequency of *filled teeth* in male subjects (Table 9), 55.3% of respondents answered that they had *no fillings or only one* filling, whereas 10.5% had six *or more* fillings.

Considering the observed results of the *OHAQ* questionnaire and the findings on the differentiation of subjects with a different *self-reported oral status* in terms of their oral hygiene activities, especially in the female subsample, we wanted to increase the practical possibilities of this questionnaire; hence, an additional analysis was conducted. Specifically, the participating dental experts were asked to determine two *cut-off* values for the *OHAQ sum* variable (*lower* and *upper* cut-off limit), referred to as the two *OHAQ index* criteria. On the basis of these criteria, the following was determined: subjects with the *OHAQ sum* result lower than the *lower limit* could possibly be invited for an urgent examination at the DMD; subjects with the *OHAQ sum* result lower than the *upper limit* could possibly be recommended to visit the DMD of their choice or given a specific recommendation related to the *lowest* expressed OHAQ scale; subjects with the *OHAQ sum* result higher than the *upper limit* could possibly be given a confirmation of their *excellent* and *high* oral hygiene activities and habits. After a discussion within the dental expert group, two cut-off values were agreed upon: 11.00 as the *lower* cut-off value and 14.00 as the *upper* cut-off value.

In the male and female subsample, two categorical variables were compared. For the first variable, the *OH type*, OHAQ measures were used for classification into four different OH types, and, for the second variable, the *OHAQ index*, only one measure (*OHAQ sum*) was used for classification into three groups. Of course, the association of these measures was expected to be significant and positive, and the *effect size* of that association was expected to be *very large*. The results of the analysis are presented in Table 10.

The correlation analysis between the two categorical variables (*OH types* and *OHAQ index*) showed an *extremely high* correlation, which was to be expected since these are two different methods of classifying subjects into *types/groups.* Above the shaded fields on the diagonal of Table 10 are those subjects who may have been classified ‘incorrectly’ or ‘poorly’ (2.5% of females and 21% of males), as, for them, the *OHAQ index* classification was *more rigorous* than the classification of *OH types*. Below the shaded fields on the diagonal of Table 10 are those subjects who may have been classified ‘incorrectly’ or ‘poorly’ (15.5% % of females and 3.2% of males), as, for them, the *OH types* classification was *more rigorous* than the classification of *OHAQ index*.

## 4. Discussion

This study identified several significant findings: the outcome of this study is a questionnaire package that can assist clinicians in predicting students’ oral health activities by assessing some of their habits and attitudes through a questionnaire; the latent structure of *oral health activities* perceived by university students was determined, and those dimensions can be measured with the scales of the OHAQ questionnaire; female and male subsamples differed in the OHAQ questionnaire measures; for both female and male subsamples, different *OH types* of activities measured with the OHAQ questionnaire were identified, showing characteristic behaviours of certain clusters of respondents; the OHAQ measures were associated to some *self-reported oral status* measures, which additionally confirmed their initial validity; this study also raised several research questions and objectives that need to be explored in the future, but the construction of satisfactory *oral health activity* measures set the stage for conducting such research.

### 4.1. Dimensions of Oral Health Activities

Oral health activity items were found to project onto five latent dimensions (BOHA, ODMD, ROTB, FLOSS, and ADOH). Of the five constructed scales, four were found to have *satisfactory* metric properties. The ADOH scale with unsatisfactory metric properties (insufficient *reliability*) was excluded from the study and should be reviewed in more detail in future studies. The identified dimensions of oral health activities may provide directions for integrating and planning activities, as well as coordinating oral health prevention efforts, especially in the young population. Furthermore, the questionnaire could be a better instrument of choice for obtaining information about individual characteristics that are not found in clinical examinations, such as daily activities related to oral health [24]. In the literature, there are few examples of questionnaires that examine oral health attitudes and behaviours, and the most frequently used questionnaire is the HU-DBI developed by Kawamura [13,14,15,16,17,18], which focuses only on the tooth brushing technique and pays little attention to other oral health activities. From a clinical point of view, the OHAQ can be used as a guidance that will help a dentist diagnose a patient’s oral health profile. Overall, the recorded data demonstrate that the OHAQ scales have satisfactory metric characteristics and are able to measure oral health activities by four different scales, which show metric characteristics of reliability and homogeneity (their internal consistency varied from *conditionally satisfactory* to *good*). Furthermore, the data confirmed a satisfactory sensitivity of the four scales.

In both gender subsamples, significant and positive associations were found among the obtained values between all four OHAQ scales. The correlation coefficients were significant but generally *low*. This finding indicates that each of the variables from the OHAQ questionnaire represents and explains only a *very small* portion of the common variance in the subsamples (from 4.4% to 16%) and that OHAQ scales measure relatively different areas of oral health activities. Associations between the OHAQ scales and the overall OHAQ sum result were positively significant and high. This finding shows that all OHAQ measures contribute to the overall measure, confirming the existence of a wide construct of oral health activities, and demonstrating the usefulness of the OHAQ sum result.

### 4.2. Frequencies and Gender Differences of Oral Health Activities

The results obtained on the total sample in this study of the four OHAQ measures are briefly presented and interpreted. More than 54% of students in Split brush their teeth for at least 3 min, change their toothbrush every 3 months, use fluoride toothpastes, and are confident that they know how to brush their teeth properly. Previous research in Croatia has shown that most students believe they know how to brush their teeth properly (78.8%), have received instructions on proper care (74.2%), and have been using the same toothbrush for less than 3 months (48.3%) [25]. Almost 70% of Croatian adolescents brush their teeth at least twice a day, and 30% of them use oral hygiene aids in addition to brushing, although 80% do not floss at all [5]. This study shows that about 70% of students at the University of Split use a safe approach to care, i.e., they apply measures that can contribute to better and safer care of the oral cavity and teeth, such as *never skip evening brushing*—74% or *use fluoride toothpaste*—41%. Preferably, toothpaste should contain fluoride, which promotes remineralisation and slows demineralisation of tooth structure, and antibacterial agents [26]. Regular check-ups, *at least once a year*, and *tartar cleaning* were not among the characteristics of the subjects in this study. Sociodemographic factors determine whether Chilean adolescents in Santiago go for a dental check-up, and adolescents who do not go for annual check-ups are mostly male, rarely brush their teeth, and have a low-income father and mother with only primary school education [27]. Furthermore, clinical examination revealed their poor oral health and that they were more likely to attend poorer schools. Research conducted in Mexico on dental students suggests that the main reason for poor prevention practices in dentistry is the lack of involvement of dentists [28]. In addition, efforts should be made to create positive attitudes towards oral health prevention and identify the need for education and training of oral health prevention experts [29].

In the present sample, gender differences were found, and females showed better results than males on all OHAQ scales. Compared to males, females brushed their teeth more often, showed a better basic approach to oral hygiene, used dental floss, and attended dental check-ups more often. However, a national survey-based study on Estonian dental students’ oral health-related *knowledge*, *attitudes,* and *behaviours* showed that there was no single significant difference between females and males in any item. Nevertheless, females had higher assessments than males in items *brushing without toothpaste* and *post-brushing checking* [17]. Moreover, a study on the Croatian adolescents sample in cities revealed that oral health was influenced by various demographic and social factors, including gender, and the study confirmed previous findings that oral hygiene-related attitudes and behaviours in adolescents are gender-related (e.g., females attached more importance to oral health and hygiene than males) [5]. A study conducted on Turkish dental students revealed a significant difference between improvement in oral health behaviours and oral hygiene habits and increasing educational level, and dental hygiene was better in females than in males [30]. Male health science students in Kuwait showed good oral health knowledge but poor practice and habits compared to their female peers [31]. Kuwaiti female students brushed their teeth more often than male students, were more aware and concerned about oral health problems, and invested more in oral hygiene than male students [32].

It can be concluded that the observed gender differences in oral health activities and behaviours among students from this study additionally confirm the literature findings [5,30,31,32].

### 4.3. Types of Oral Health Activities

In both males and females, four different types of *oral health* activities were identified: *excellent*, *good*, *satisfactory*, and *poor OH* type. The structure of the identified OH types in the female subsample and the male subsample had many similarities and shared some basic characteristics and relationships between the scores on OHAQ scales. Although the OH types in the female and male subsample had identical names, there were substantial differences between them in the frequency or in the ‘extent’ of certain OH activities. The differences between female and male OH types could be attributed to the previously determined gender differences on all OHAQ scales. These differences are consistent with other findings in the literature that detected gender differences, with either different behaviours or different frequencies of certain behaviours [5,28,29,30,31].

Furthermore, significant findings were the relative frequencies and basic characteristics of the identified OH types. In the female subsample, the relative frequencies of OH types were *excellent*—19.6%, *good*—27.3%, *satisfactory*—27.3%, and *poor*—25.7%. In the male subsample, the relative frequencies of OH types were *excellent*—9.1%, *good*—32.4%, *satisfactory*—30.1%, and *poor*—28.3%. The *excellent* OH type in the female subsample was about twice as numerous as the same OH type in the male subsample, and relative values of the other OH types were less pronounced. Given the basic characteristics of the identified OH types, it was justified to assume that only the members of the two *excellent* types would have high-quality OH care in all the areas needed (*regularity of tooth brushing*, *application of basic activities*, *orientation to DMD*, and *use of dental floss*). All other OH types in both subsamples need certain improvements and changes in their behaviours, to a greater or lesser extent, to achieve excellent OH care. For example, members of the *good* OH type should increase the *frequency of flossing* despite the very good characteristics of *regular tooth brushing* and *basic OH activities*. Members of the *satisfactory* OH type should increase the frequency of all their OH activities, *orientation to DMD*, and other activities measured by the OHAQ scales. Members of the *poor* OH type should significantly change their behaviours related to performing OH activities, as their behaviours were very undesirable and of poor quality.

Comparison of the identified OH types in the female subsample to the *self-reported oral status* measures revealed significant differences between different OH type groups in two oral status measures: *toothache* and the number of *filled teeth*. It was found that increased quality of OH activities was associated with an increased *number of filled teeth*. Thus, females with a better and preferred OH type had a higher *number of filled teeth*. This was opposite to the expectation expressed at the beginning of this study that a higher number could objectively represent poor and less desirable characteristics of individual oral health status, but it was indeed very understandable and easy to interpret. Females who take better care of their OH are also more likely to go for regular DMD check-ups and take the necessary measures (such as *dental fillings*) to maintain their oral health and prevent further discomfort or impairment. It was found that increased quality of OH activities was associated with a lower incidence of *toothache*. Females belonging to the least desirable, *poor* OH type had an incidence of *toothache* higher than all other females. In the male subsample, differences were found between OH type groups in some *self-reported oral status* variables. Males with the *excellent* OH type had the lowest incidence of *toothache* of all the OH types. Males with the *satisfactory* OH type had the highest frequency of *use of analgesics* for toothache of all the OH types. Considering the characteristics of the *satisfactory* OH type, they might be advised to change their behaviour so that, instead of taking analgesics frequently, they focus on brushing their teeth more often and visit the DMD more frequently.

Lastly, it is reasonable to argue that the OH types identified by OHAQ questionnaire in the male and female subsamples can allow a better designing, adapting, or modulating of future education or prevention programs to a specific and targeted part of the adolescent population. For example, students (or other users of the questionnaire) may be directed or invited to different educational programs solely on the basis of questionnaire classification or the determined ‘membership’ in a particular OH gender type.

### 4.4. Self-Reported Oral Health Status and OHAQ Measures

*Self-reported oral health status* showed that male and female students had, on average, <1 *extracted tooth*, <3 *filled teeth*, and <2 *treated teeth*. As expected, higher incidences of the *extracted teeth*, *filled teeth*, and *treated teeth* had significantly lower frequencies among students. Furthermore, both female and male students rated the frequency of *toothache* and the *use of analgesics* and *antibiotics* for their teeth as *low*. A study on self-reported oral health status stated that the *behaviour*, *consciousness*, and *oral health status* of medical and dental students were not optimistic [33]. A study on Iraqi dental students showed that they had quite good behaviour and attitude toward oral health. However, the authors believed that additional focus needs to be placed on the anticipatory and behavioural aspects of oral self-help practice [34].

A review of the literature on the validity of *self-reported oral health* measures revealed that the findings are somewhat contradictory and differ in terms of variables used in individual studies. Findings that indicate that such measures have poor validity mainly relate to the variables that measure *assessment of dental caries* [35], *specific periodontal variables* [35,36,37], and *normative dental treatment needs* [38]. However, there is evidence that *self-reported oral health status* measures accurately provide *number of teeth* [35,36,39,40,41,42,43], *presence of fillings* [35], *use of dental prosthesis* [36], *periodontal disease* [39,44,45,46], *orofacial pain* [47], *root canal treatment* (RCT) [35,37], and *orthodontic and endodontic needs* [38]. These validity-supporting authors also stated the following reasons and benefits for using *self-reported oral health* measures: for great cost and time savings [35,46], as a valid method to determine the number of teeth in national health surveys [39], as accurate diagnostics for predicting orthodontic and endodontic needs [38], as a valid reflection of the clinical status [40], as a valuable tool for epidemiological studies and surveillance of *periodontal health* in the adult population [36,45], and as possible guidance for people in making improvements in their lifestyle [48]. According to the previous claims, we conclude that it was reasonable to assume that *self-reported*
*oral health* measures could be a useful and good basis for the implementation of a rapid and rough classification of students’ oral health status, as well as for the initial validation of the OHAQ measures. Of course, in future studies involving the OHAQ questionnaire and for further validation of the questionnaire, it is recommended to use oral status measures determined by clinical examination.

On the basis of the present findings, *oral health activities* are related to the frequency of *toothache.* A regular level of *basic oral hygiene activities* appears to be related to a lower incidence of *toothache* in the entire population, independently of gender. In females, the orientation to a DMD, the use of dental floss, and the regularity of tooth brushing are also significantly correlated with the frequency of toothache. These findings are consistent with other data from the literature [49,50]. From a more general point of view, it was expected that the oral health activities and behaviour scales presented in this study could differentiate subjects with insufficient oral health activities and high-risk behaviour for toothache. The scales showed good discriminant predictive value in differentiating subjects according to their frequency of toothache. Regarding the number of *filled teeth*, *basic oral health activities* were higher in females with *six*
*or more* filled teeth than in those with fewer fillings. Furthermore, the *orientation to a DMD* and the *use of dental floss* were more frequent in female students with *six or more* tooth fillings. A possible explanation for this observation is that the experiences of female students with *six*
*or more filled teeth* probably reflect some behavioural alterations and more frequent and higher-quality practices of *basic oral hygiene activities* in comparison to females with *only one or no filled* teeth. It is reasonable to conclude that these findings further confirm good *discriminant validity* of the OHAQ in relation to *self-reported oral status* measures (e.g., *frequency of toothache* and *number of filled teeth*).

### 4.5. Usage of the OHAQ Index Criteria

By using the *OH types* classification, all subjects were classified into four different OH types (*excellent*, *good*, *satisfactory*, and *poor*), and, by using the expert-established *OHAQ index* criteria, all students were divided into three different groups (with *high*, *moderate*, or *low* overall OHAQ results). Considering these two different classifications of subjects, we (the authors of the study) noticed a “problem” in the central part of the classifications, representing the *good* and *satisfactory* OH types and the *moderate OHAQ index* group. A large percentage of females were below the diagonal (15.5%), and a small percentage were above the diagonal (2.5%) of these two classifications. Exactly the opposite, a large percentage of males were above the diagonal (21%), and a small percentage were below the diagonal (3.2%) of these two classifications. From these observed differences in the classifications of females and males, it can be concluded that the *OHAQ index* criteria established by the experts were ‘*permissive*’ for the female subsample and ‘*severe*’ for the male subsample. As an outcome of gender differences in OHAQ measures and of these expert-established criteria, male subjects tended to be assigned to a lower *OHAQ index* group, whereas female subjects tended to be assigned to a higher *OHAQ index* group.

Nevertheless, we believe that the *OHAQ index* criteria were well established and that the observed discrepancy was due to the previously determined gender differences in all the OHAQ measures. Of course, it could be recommended for these *OHAQ index* criteria to be tested further in practice so that additional specific expert-defined criteria can be developed for each of the four OHAQ measures. Systematic reviews have shown that oral health education has a positive impact on the health, knowledge, and practical behaviour of children and adolescents [51,52]. However, the authors did not find a similar *index* or *criterion* for a rapid and rough classification of adolescents.

### 4.6. Limitations of the Study

This study is a cross-sectional study that based its findings on participants’ self-reports. Moreover, one limitation might be the way the OHAQ construct was measured. The OHAQ measures had only a *conditionally satisfactory* level of reliability, and the process of their advancement has to be continued. It is necessary to increase the number of items in the scales in future studies. Furthermore, it is necessary to reconsider the measurement of the ADOH subscale in a more detailed way. Perhaps a possibility of using some particular items from that scale as a single item construct should be considered. Although the literature confirms the satisfactory validity of *self-reported oral health* measures, we think it is necessary to perform an additional validation of OHAQ measures through a clinical examination of subjects. In this research, the implementation of clinical examination of subjects was not conducted due to the application of a developing questionnaire entirely in classrooms at the end of a lecture, and the filling out of the questionnaire was carried out anonymously. Furthermore, for future research, it will be necessary to verify the proposed categorisation based on the *OHAQ index* criteria, as well as provide additional criteria for each OHAQ measure. In addition to completing the questionnaire, it is necessary to invite students to an examination by a DMD to determine their oral status and to invite them to participate in educational OH programs. It is also possible to check the relationships between the OHAQ measures and some other possibly valuable measures, such as measures of *body image (general teeth satisfaction)*, *OH attitudes*, *sources of OH behaviours*, *self-esteem*, and the traits of *optimism* and *hope*.

## 5. Conclusions

The *construct validity* of the OHAQ questionnaire was found to be *good*, considering the *satisfactory* metric properties of its scales. Good *construct validity* manifested itself in several important findings: the scales of the OHAQ questionnaire measured relatively independent domains of OH yet contributed significantly and to a large extent to the explanation of the overall measure of OH activities; the questionnaire scales showed good *discriminant validity*, revealing both gender differences and differences related to specific elements of subjects’ *self-reported oral status* measures (e.g., *frequency of toothache* and the *number of filled teeth*); on the basis of the OHAQ measures, it was possible to identify different OH types in both gender subsamples, with significant differences found in their OH behaviours and habits. It is reasonable to conclude that by measuring all four OHAQ measures, it could be possible to get a good and broad insight into OH activities and behaviours of the subjects. Furthermore, there is a need for additional validation in future research of OHAQ measures through a clinical examination of subjects, which was not performed in the present study because of the anonymous administration of the OHAQ questionnaire among students.

The OHAQ questionnaire accomplished the set goal, which was to provide a tool for quick, short, and good classification of subjects based on their OH activities and behaviours. It is recommended to be used both in future research (while improving the metric properties of the OHAQ scales) and in practice, both for rapidly classifying subjects and for measuring the possible impact of oral care educational programs implemented in pre-, post-, and possible follow-up tests.

## Figures and Tables

**Figure 1 ijerph-19-05556-f001:**
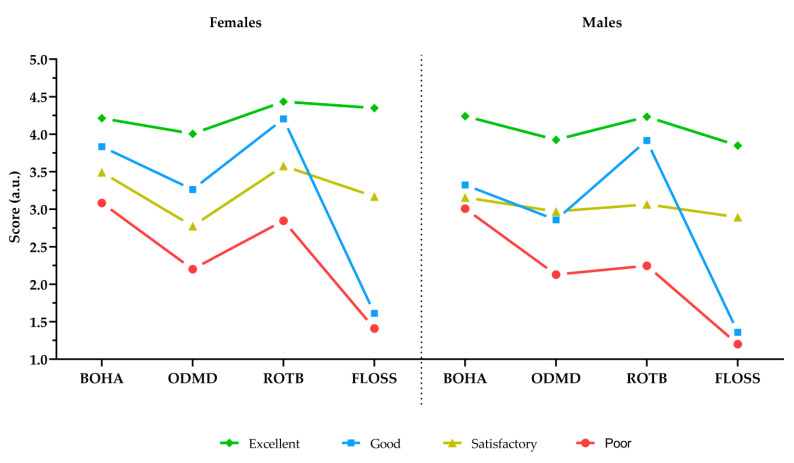
Female and male *OH types*.

**Table 1 ijerph-19-05556-t001:** Factor analysis of the *Oral Health Activities Questionnaire* items.

Item	Component *				
	1	2	3	4	5
I am certain I brush my teeth properly	0.68	0.01	0.17	0.03	−0.07
I use small, circular rotations while tooth brushing	0.63	0.14	−0.01	0.12	−0.06
I replace my toothbrush every 3 months	0.60	0.16	0.04	0.10	0.20
I use fluoride toothpaste	0.48	0.13	0.11	0.00	0.39
I brush my teeth for at least 3 min	0.42	0.17	0.23	0.16	−0.09
I brush my tongue when tooth brushing	0.43	−0.12	0.16	0.18	0.33
I have professional dental scaling at least once per year	0.01	0.74	0.03	0.28	0.16
I have professional dental scaling regularly	0.05	0.73	0.08	0.35	0.15
I visit a doctor of dental medicine twice per year for regular check-ups	0.20	0.60	0.27	0.04	0.04
I visit a doctor of dental medicine mostly for regular check-ups	0.38	0.53	0.07	−0.16	0.06
I brush my teeth at least three times per day	0.10	0.11	0.81	0.13	0.17
I brush my teeth after each meal	0.03	0.15	0.77	0.14	0.17
I never omit evening tooth brushing	0.27	0.09	0.64	0.00	−0.18
I use dental floss at least once per day	0.18	0.18	0.15	0.82	0.05
I use dental floss often during the day	0.09	0.22	0.10	0.81	0.14
I use interdental brushes	0.02	0.17	0.13	−0.06	0.66
I use an electric toothbrush	−0.20	0.14	−0.13	0.05	0.59
When I brush my teeth, I intentionally massage the gingiva	0.17	0.00	0.07	0.27	0.50
I use mouth rinse	0.27	−0.02	0.07	0.30	0.36
EIGEN	2.27	2.00	1.93	1.85	1.66
% VAR	12.0	10.5	10.9	9.7	8.7
Total %	51.1				

Notes: * factor saturation; EIGEN—eigen value, characteristic variance of component; %—percentage of variance explained by component; Total %—total percentage of explained variance.

**Table 2 ijerph-19-05556-t002:** Oral Health Activities Questionnaire scales.

Basic Oral Hygiene Activities Scale					
Items	EIGEN	% VAR	Alpha	FS	Mean ± SD
	2.16	36.1	0.64		
I replace my toothbrush every 3 months				−0.67	3.48 ± 1.30
I am certain I brush my teeth properly				−0.65	3.81 ± 1.03
I use small, circular rotations while tooth brushing				−0.63	3.53 ± 1.17
I use fluoride toothpaste				−0.60	3.22 ± 1.34
I brush my teeth for at least 3 min				−0.54	3.55 ± 1.13
I brush my tongue when tooth brushing				−0.50	3.44 ± 1.37
**Orientation to Dental Medicine Doctor Scale**					
Items	EIGEN	% VAR	Alpha	FS	Mean ± SD
	2.06	51.5	0.68		
I have professional dental scaling regularly				0.83	2.34 ± 1.26
I have professional dental scaling at least once per year				0.81	2.64 ± 1.42
I visit a doctor of dental medicine twice per year for regular check-ups				0.66	3.47 ± 1.34
I visit a doctor of dental medicine mostly for regular dental check-ups				0.54	3.27 ± 1.27
**Regularity of Tooth Brushing Scale**					
Items	EIGEN	% VAR	Alpha	FS	Mean ± SD
	1.84	61.3	0.68		
I brush my teeth at least three times per day				0.85	3.57 ± 1.27
I brush my teeth after each meal				0.82	2.98 ± 1.26
I never omit evening tooth brushing				0.67	4.13 ± 1.15
**Use of Dental Floss Scale**					
Items	EIGEN	% VAR	Alpha	FS	Mean ± SD
	1.69	84.3	0.81		
I use dental floss often during the day				0.92	2.28 ± 1.27
I use dental floss at least once per day				0.92	2.42 ± 1.43
**Additional and Detail Oral Hygiene Activities Scale**					
Items	EIGEN	% VAR	Alpha	FS	Mean ± SD
	1.47	36.7	0.42		
I use interdental brushes				−0.65	2.61 ± 1.31
When I brush my teeth, I intentionally massage the gingiva				−0.64	3.06 ± 1.32
I use mouth rinse				−0.58	3.06 ± 1.42
I use an electric toothbrush				−0.55	1.70 ± 1.15

Notes: EIGEN—eigen value, component variance; % VAR—percentage of variance of component; Alpha—Cronbach’s alpha coefficient; FS—factor saturation by component.

**Table 3 ijerph-19-05556-t003:** Sensitivity and gender differences of OHAQ scales.

Variable	Female Students (*N* = 439)					Male Students (*N* = 219)					*t*-Test	*p*
	Mean ± SD	MED	SKEW	KURT	K–S D	Mean ± SD	MED	SKEW	KURT	K–S D		
Basic oral hygiene activities	3.62 ± 0.74	3.50	−0.01	−0.46	0.08 *	3.27 ± 0.66	3.17	0.33	−0.02	0.11 *	6.01	<0.001
Orientation to DMD	3.00 ± 0.96	3.00	0.23	−0.50	0.09 *	2.78 ± 0.89	2.75	0.19	−0.15	0.07	2.82	0.005
Regularity of tooth brushing	3.73 ± 0.92	4.00	−0.60	−0.05	0.14 *	3.22 ± 0.96	3.00	−0.03	−0.56	0.09	6.65	<0.001
Use of dental floss	2.52 ± 1.28	2.50	0.47	−0.87	0.14 *	2.00 ± 1.08	2.00	0.85	−0.28	0.21 *	5.15	<0.001
OHAQ Sum	12.88 ± 2.81	12.58	0.24	−0.24	0.06	11.27 ± 2.49	11.00	0.48	0.44	0.05	7.16	<0.001

Notes: DMD, dental medicine doctor; mean ± SD—arithmetic mean and standard deviation; MED—median; SKEW—coefficient of asymmetry of distribution; KURT—coefficient of kurtosis of distribution; K–S D—Kolmogorov–Smirnov *goodness-of-fit* test; * significant K–S D test coefficient; *t*-test—*t*-test coefficient; *p*—significance of the *t*-test coefficient.

**Table 4 ijerph-19-05556-t004:** Correlations between the OHAQ measures.

**Variable**	**Females (*N* = 439)**			
	**Basic Oral** **Hygiene Activities**	**Orientation** **to DMD**	**Regularity of** **Tooth Brushing**	**Use of** **Dental Floss**
Basic oral hygiene activities	1.00	0.39 **	0.37 **	0.32 **
Orientation to DMD	0.39 **	1.00	0.35 **	0.40 **
Regularity of tooth brushing	0.37 **	0.35 **	1.00	0.28 **
Use of dental floss	0.32 **	0.40 **	0.28 **	1.00
OHAQ Sum	0.66 **	0.74 **	0.67 **	0.77 **
**Variable**	**Males (*N* = 219)**			
	**Basic Oral** **Hygiene Activities**	**Orientation** **to DMD**	**Regularity of** **Tooth Brushing**	**Use of** **Dental Floss**
Basic oral hygiene activities	1.00	0.36 **	0.30 **	0.29 **
Orientation to DMD	0.36 **	1.00	0.29 **	0.39 **
Regularity of tooth brushing	0.30 **	0.29 **	1.00	0.21 *
Use of dental floss	0.29 **	0.39 **	0.21 *	1.00
OHAQ Sum	0.64 **	0.74 **	0.66 **	0.73 **

Notes: * *p* < 0.05; ** *p* < 0.001.

**Table 5 ijerph-19-05556-t005:** *OH types* for female and male students.

**Variable**	**Female OH Types**				**F**	** *p* **
	** *Excellent* ** **(*N* = 86)**	** *Good* ** **(*N* = 120)**	** *Satisfactory* ** **(*N* = 120)**	** *Poor* ** **(*N* = 113)**		
Basic oral hygiene activities	4.22 ± 0.60	3.83 ± 0.63	3.49 ± 0.61	3.08 ± 0.65	60.52	<0.001
Orientation to DMD	4.01 ± 0.82	3.26 ± 0.78	2.78 ± 0.70	2.20 ± 0.61	109.15	<0.001
Regularity of tooth brushing	4.43 ± 0.53	4.21 ± 0.60	3.58 ± 0.75	2.85 ± 0.82	111.81	<0.001
Use of dental floss	4.35 ± 0.69	1.61 ± 0.54	3.17 ± 0.56	1.41 ± 0.51	587.78	<0.001
OHAQ Sum	17.00 ± 1.37	12.92 ± 1.41	13.02 ± 1.13	9.54 ± 1.26	543.61	<0.001
**Variable**	**Male OH Types**				**F**	** *p* **
	** *Excellent* ** **(*N* = 20)**	** *Good* ** **(*N* = 71)**	** *Satisfactory* ** **(*N* = 66)**	** *Poor* ** **(*N* = 62)**		
Basic oral hygiene activities	4.24 ± 0.53	3.32 ± 0.59	3.16 ± 0.53	3.01 ± 0.61	24.42	<0.001
Orientation to DMD	3.93 ± 0.89	2.86 ± 0.85	2.97 ± 0.61	2.13 ± 0.69	33.38	<0.001
Regularity of tooth brushing	4.23 ± 0.77	3.92 ± 0.65	3.07 ± 0.63	2.25 ± 0.57	95.93	<0.001
Use of dental floss	3.85 ± 0.73	1.36 ± 0.47	2.89 ± 0.68	1.20 ± 0.36	213.85	<0.001
OHAQ Sum	16.25 ± 1.51	11.46 ± 1.39	12.09 ± 1.27	8.59 ± 1.17	192.68	<0.001

Notes: Data are presented as the mean ± SD; F—analysis of the variance coefficient; *p*—significance of the ANOVA coefficient.

**Table 6 ijerph-19-05556-t006:** Descriptive statistics of *self-reported oral status* measures.

Variable	Female Students				Male Students			
	Mean ± SD	Median	Min	Max	Mean ± SD	Median	Min	Max
**Tooth extraction**	0.69 ± 1.16	0	0	8	0.49 ± 1.05	0	0	5
**Filled teeth**	2.92 ± 2.30	2	0	9	2.27 ± 2.27	1	0	9
**Root canal treatment**	1.31 ± 1.67	1	0	9	1.00 ± 1.16	1	0	6
**Toothache**	2.08 ± 0.89	2	1	5	2.02 ± 0.91	2	1	5
**Use of analgesics**	1.46 ± 0.71	1	1	5	1.41 ± 0.68	1	1	5
**Use of antibiotics**	1.35 ± 0.67	1	1	5	1.26 ± 0.57	1	1	4
**Variable**	**Yes answer**		**No answer**		**Yes answer**		**No answer**	
	**Frequency**	**%**	**Frequency**	**%**	**Frequency**	**%**	**Frequency**	**%**
**Malocclusion**	122	27.8	317	72.2	57	26.0	162	74.0
**Orthodontic treatment**	200	45.6	239	54.4	82	37.4	137	62.6
**Dental crown or veneer**	25	5.7	414	94.3	12	5.5	207	94.5
**Prosthetic treatment**	10	2.3	429	97.7	6	2.7	213	97.3

Notes: Mean ± SD—arithmetic mean ± standard deviation; Min—minimum result; Max—maximum result; %—percentage.

**Table 7 ijerph-19-05556-t007:** ANOVA of *OH type* groups by *self-reported oral status* measures.

**Variable**	**Female OH Types**				**F**	** *p* **
	** *Excellent* ** **(*N* = 86)**	** *Good* ** **(*N* = 120)**	** *Satisfactory* ** **(*N* = 120)**	** *Poor* ** **(*N* = 113)**		
**Tooth extraction**	0.81 ± 1.23	0.57 ± 1.02	0.66 ± 1.15	0.77 ± 1.27	0.92	0.43
**Filled** **teeth**	3.57 ± 2.76	3.09 ± 2.23	2.79 ± 2.25	2.39 ± 1.91	4.75	0.003
**Root canal treatment**	1.20 ± 1.75	1.49 ± 1.86	1.25 ± 1.55	1.25 ± 1.54	0.70	0.55
**Toothache**	1.87 ± 0.84	2.05 ± 0.86	2.02 ± 0.81	2.34 ± 0.99	5.08	0.002
**Use of analgesics**	1.45 ± 0.70	1.42 ± 0.63	1.42 ± 0.71	1.54 ± 0.81	0.66	0.57
**Use of antibiotics**	1.36 ± 0.61	1.33 ± 0.64	1.34 ± 0.64	1.38 ± 0.76	0.11	0.95
**Malocclusion**	0.22 ± 0.42	0.33 ± 0.47	0.26 ± 0.44	0.29 ± 0.46	1.02	0.38
**Orthodontic treatment**	0.47 ± 0.50	0.49 ± 0.50	0.43 ± 0.50	0.43 ± 0.50	0.37	0.77
**Variable**	**Male OH types**				**F**	** *p* **
	** *Excellent* ** **(*N* = 20)**	** *Good* ** **(*N* = 71)**	** *Satisfactory* ** **(*N* = 66)**	** *Poor* ** **(*N* = 62)**		
**Tooth extraction**	0.40 ± 0.75	0.45 ± 0.97	0.58 ± 1.12	0.47 ± 1.16	0.24	0.87
**Filled** **teeth**	2.10 ± 2.07	2.14 ± 2.06	2.61 ± 2.47	2.11 ± 2.35	0.69	0.56
**Root canal treatment**	0.75 ± 1.25	1.11 ± 1.32	1.08 ± 1.19	0.87 ± 0.88	0.88	0.45
**Toothache**	1.55 ± 0.94	2.04 ± 0.92	2.05 ± 0.88	2.11 ± 0.91	2.04	0.11
**Use of analgesics**	1.25 ± 0.44	1.32 ± 0.55	1.59 ± 0.86	1.35 ± 0.63	2.49	0.06
**Use of antibiotics**	1.15 ± 0.37	1.17 ± 0.38	1.41 ± 0.74	1.26 ± 0.57	2.40	0.07
**Malocclusion**	0.15 ± 0.37	0.30 ± 0.46	0.21 ± 0.41	0.31 ± 0.46	1.07	0.36
**Orthodontic treatment**	0.25 ± 0.44	0.42 ± 0.50	0.33 ± 0.48	0.40 ± 0.49	0.90	0.44

Notes: Data are presented as the mean ± SD; F—analysis of the variance coefficient; *p*— significance of the ANOVA coefficient.

**Table 8 ijerph-19-05556-t008:** Analysis of variance of the *OH status* groups for female students.

**Variable**	**Frequency of *toothache* groups**				**F**	** *p* **
	** *Never* ** **(*N* = 113)**	** *Very Rarely* ** **(*N* = 219)**	** *Rarely* ** **(*N* = 69)**	** *Sometimes +* ** ***Often* (*N* = 38)**		
Basic oral hygiene activities	3.77 ± 0.78	3.66 ± 0.71	3.38 ± 0.71	3.43 ± 0.71	5.11	0.002
Orientation to DMD	3.15 ± 1.00	3.02 ± 0.96	2.97 ± 0.89	2.49 ± 0.81	4.63	0.003
Regularity of tooth brushing	3.86 ± 0.89	3.77 ± 0.94	3.52 ± 0.88	3.49 ± 0.84	2.92	0.034
Use of dental floss	2.80 ± 1.34	2.50 ± 1.26	2.38 ± 1.22	2.12 ± 1.21	3.43	0.017
**Variable**	**Number of *filled teeth* groups**				**F**	** *p* **
	**0–1** **(*N* = 170)**	**2–3** **(*N* = 106)**	**4–5** **(*N* = 111)**	**6 *or more* ** **(*N* = 52)**		
Basic oral hygiene activities	3.45 ± 0.69	3.67 ± 0.77	3.74 ± 0.71	3.87 ± 0.78	6.35	<0.001
Orientation to DMD	2.99 ± 0.97	2.95 ± 0.96	2.92 ± 0.97	3.34 ± 0.88	2.62	0.050
Regularity of tooth brushing	3.70 ± 0.98	3.74 ± 0.95	3.70 ± 0.81	3.89 ± 0.85	0.66	0.58
Use of dental floss	2.51 ± 1.27	2.31 ± 1.18	2.51 ± 1.35	3.01 ± 1.26	3.54	0.015

Notes: Data are presented as the mean ± SD; F—analysis of the variance coefficient; *p*—significance of the ANOVA coefficient.

**Table 9 ijerph-19-05556-t009:** Analysis of variance of the *OH status* groups for male students.

**Variable**	**Frequency of *toothache* groups**				**F**	** *p* **
	** *Never* ** **(*N* = 67)**	** *Very Rarely* ** **(*N* = 99)**	** *Rarely* ** **(*N* = 39)**	** *Sometimes + Often* ** **(*N* = 14)**		
Basic oral hygiene activities	3.50 ± 0.69	3.16 ± 0.65	3.16 ± 0.54	3.20 ± 0.68	4.16	0.007
Orientation to DMD	2.93 ± 1.04	2.76 ± 0.83	2.64 ± 0.65	2.70 ± 1.11	0.97	0.41
Regularity of tooth brushing	3.42 ± 1.04	3.18 ± 0.87	3.00 ± 0.93	3.12 ± 1.19	1.76	0.16
Use of dental floss	2.10 ± 1.16	1.97 ± 1.07	1.99 ± 0.98	1.79 ± 1.05	0.41	0.75
**Variable**	**Number of *filled teeth* groups**				**F**	** *p* **
	**0–1** **(*N* = 121)**	**2–3** **(*N* = 43)**	**4–5** **(*N* = 32)**	**6 *or more* ** **(*N* = 23)**		
Basic oral hygiene activities	3.22 ± 0.62	3.48 ± 0.67	3.31 ± 0.72	3.06 ± 0.70	2.48	0.06
Orientation to DMD	2.83 ± 0.91	2.66 ± 0.90	2.88 ± 0.87	2.66 ± 0.86	0.63	0.60
Regularity of tooth brushing	3.22 ± 0.96	3.40 ± 0.92	3.11 ± 0.97	3.00 ± 1.03	1.05	0.37
Use of dental floss	2.00 ± 1.12	2.02 ± 1.03	2.05 ± 1.12	1.93 ± 0.91	0.05	0.98

Notes: Data are presented as the mean ± SD; F—analysis of the variance coefficient; *p*—significance of the ANOVA coefficient.

**Table 10 ijerph-19-05556-t010:** Association of *OHAQ type* and *OHAQ index* variables for female and male students.

**Female students**		**OHAQ index group**			**All**	**Test of Association**	
		**High**	**Moderate**	**Low**			
**OH** **type**	Excellent	86	0	0	86	Chi-square test	529.31
	Good	26	88	6	120	df	6
	Satisfactory	29	88	3	120	*p*	<0.001
	Poor	0	13	100	113	Cramer’s V	0.78
	All	141	189	109	439	*p*	<0.001
**Male students**		**OHAQ index group**			**All**	**Test of Association**	
		**High**	**Moderate**	**Low**			
**OH** **type**	Excellent	20	0	0	20	Chi-square test	241.68
	Good	2	42	27	71	df	6
	Satisfactory	4	43	19	66	*p*	<0.001
	Poor	0	1	61	62	Cramer’s V	0.74
	All	26	86	107	219	*p*	<0.001

Notes: df—degrees of freedom; Cramer’s V—Cramer’s V effect size of the chi-square test; *p*—level of test significance.

## Data Availability

The raw data supporting the conclusions of this article may be made available by the authors, without undue reservation, upon written request.

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
