# Peer review of "Development and Initial Validation of the Oral Health Activities Questionnaire"

_ijerph, 2022, doi:10.3390/ijerph19095556_

Round 1

Reviewer 1 Report

Good job.I think this article is well-written and worth publishing in the journal.

Author Response

Dear Sirs,
the answer to your review is attached.

Reviewer 2 Report

The authors have presented an important topic in a well scientific manner. I find the results logical. However, I would suggest to change the patterns of the lines in figure 1. All are rounded, but they should be diamond, square, triangle and rounded. 

They covered almost everything including the limitations of the study, I found that  the figure 1 should be changed and suggested in the comments section. However, they could enlarge the introduction a bit with more literature review, with information such as: - adding similar studies in other fields and participants  - writing the gaps in the literature and how this manuscript will cover it.   

Author Response

(The authors gave the same response as above.)

Reviewer 3 Report

This is an interesting study on a relevant topic. The main issue is the validation using self-reported oral health measures. Although previous studies have shown that self-reports regarding very basic measures such as the number of remaining teeth are often valid, results on the number of fillings or root canal treatments are less promising, many people cannot assess this correctly. 

Although this is shortly mentioned as a study limitation in the discussion section, it is necessary to include this important limitation to the abstract and conclusion and to include literature on the validity of self-reports. 

If possible, a subsection of the repondents should be examined clinically to validate the questionnaire.

Author Response

(The authors gave the same response as above.)

Reviewer 4 Report

Thank you very much for the very interesting and important manuscript titled “Development and initial validation of the Oral Health Activities Questionnaire”. The problem of oral health habits is fundamental in patients of all ages – children and adults. The questionnaire helps know the dental hygiene habits of the populations or individuals; however, in my opinion, the dental examination of the examined students should be done as well.

Despite all, the manuscript is well written and is a source of essential information. Maybe the authors could continue the study in this group of patients,

Author Response

(The authors gave the same response as above.)

Round 2

Reviewer 3 Report

The authors have conducted a sufficient literature search on the validity of self-reports. However, some amendments are still necessary:

  • Please transfer the aspects regarding the literature on self-reports from section 2.3 to the discussion section, as this is not relevant to the method description per se, but to the critical discussion one’s own methods. I would recommend placing these aspects somewhere around l.570.
  • Please also include literature that does not support your choice of methods. You have only mentioned those aspects that support the validity of self-reports. However, some authors have found limitations to self-reports (for instance, Ref. 5 or 11 in your table). This must be critically discussed.
  • If it is not possible to clinically examine your participants, this must be acknowledged as a limitation. This is shortly mentioned in l. 573, but should also be mentioned in the conclusion and also in the abstract (!).
  • Many people will only read the abstract. However, it is not clear from the abstract that the questionnaire was validated against self-reported oral health measures. One would obviously assume that e.g. the number of fillings mentioned were actually clinically determined. Please include this information in the methods section of the abstract

In general, describing one’s methods unambiguously and acknowledging limitations is not a weakness (!) but good clinical practice and improves the quality of the research.

Author Response

Dear Sirs,
I am sending the attached answer to your 2nd round of reviews.
